# A Robust Steered Response Power Localization Method for Wireless Acoustic Sensor Networks in an Outdoor Environment

**DOI:** 10.3390/s21051591

**Published:** 2021-02-25

**Authors:** Yiwei Huang, Jianfei Tong, Xiaoqing Hu, Ming Bao

**Affiliations:** 1Key Laboratory of Noise and Vibration Research, Institute of Acoustics, Chinese Academy of Sciences, Beijing 100190, China; huangyiwei@mail.ioa.ac.cn (Y.H.); tongjianfei@mail.ioa.ac.cn (J.T.); auxqhu@gmail.com (X.H.); 2University of Chinese Academy of Sciences, Beijing 100049, China

**Keywords:** source localization, wireless acoustic sensor networks, steered response power, generalized cross-correlation

## Abstract

The localization of outdoor acoustic sources has attracted attention in wireless sensor networks. In this paper, the steered response power (SRP) localization of band-pass signal associated with steering time delay uncertainty and coarser spatial grids is considered. We propose a modified SRP-based source localization method for enhancing the localization robustness in outdoor scenarios. In particular, we derive a sufficient condition dependent on the generalized cross-correlation (GCC) waveform function for robust on-grid source localization and show that the SRP function with GCCs satisfying this condition can suppress the disturbances induced by the grid distance and the uncertain steering time delays. Then a GCC refinement procedure for band-pass GCCs is designed, which uses complex wavelet functions in multiple sub-bands to filter the GCCs and averages the envelopes of the filtered GCCs as the equivalent GCC to match the sufficient condition. Simulation results and field experiments demonstrate the excellent performance of the proposed method against the existing SRP-based methods.

## 1. Introduction

With the rapid development of communication technology and mobile computing devices, applications of wireless acoustic sensor networks (WASNs) are becoming popular in acoustic signal processing. Particularly, WASN-based sound source localization has captured researchers’ attention in the last two decades [1,2,3,4,5]. The existing methods available for passive source localization in WASNs include (1) the received energy-based approaches [6,7,8,9]; (2) the direction of arrival (DOA)-based approaches [10,11]; (3) the time of arrival (TOA)-based approaches [12]; (4) the time difference of arrival (TDOA)-based approaches [13,14,15] and (5) the steered response power (SRP)-based approaches [16,17,18,19,20,21,22].

Most methods require a pre-processing stage in which specific modalities are measured from sensor signals before the location-estimating stage. In contrast, the SRP-based approaches locate the source position or direction by maximizing the power of spatially steered filter and sum beamformer of a group of sensors and contain only one decision step in processing sensor signals to estimate location. Without information compression and disturbances resulting from partial mistakes in the front-end stage, the SRP-based solutions can usually yield more robust performance in noisy and reverberant acoustic environments. Practical implementations commonly use the generalized cross-correlation [23]-based form of the SRP function [16] to reduce computation. The methods similar to the GCC-expression of SRP function are also called a “global coherence field (GCF)” in several references [24,25].

In practice, the primary constraint of the SRP-based approaches is the time-consuming on-grid searching procedure for finding their global maximums. Hence, it has been a hot issue to reduce the computational cost for the SRP-based approaches. In [17], a stochastic region construction (SRC) method is proposed to avoid global grid searching. However, this strategy also causes information loss. In [26], a geometrically sampled grid set based on the TDOA gradient is proposed to improve the SRP performances. An alternative strategy to solve the high-cost searching problem is adopting some adaptive SRP functions regarding the grid resolution to apply a coarse or a hierarchical searching. In [27], the authors use the low-frequency component of GCC for coarse grid resolution and the high-frequency component for fine grids in the SRP-based DOA estimation. In [28], the authors adopt a Gaussian low-pass filter to the GCC for coarse grids. For full-band signals, a similar kind of modification is proposed both in microphone arrays [29] and WASNs [18,19], respectively, in which the spatial spectrum of a given grid is calculated from the sum of the phase-transform weighted GCCs (GCC-Phase Transform (PHAT)s) within a time window containing the TDOA values in the volume surrounding the grid, instead of the original GCC-PHAT in the SRP function.

The SRP-based approaches can provide a robust solution in DOA estimation and source localization tasks in confined spaces. However, they could lose their robustness in an outdoor WASN scenario due to the synthetic effect of the following factors. (1) Grid size, since the monitoring area in outdoor cases may become much more extensive than the area of indoor applications, and the proper searching grids would be much coarser (e.g., meter-level grids outdoors compared with centimeter-level grids indoors). (2) Steering time delay uncertainty; in the classical SRP-based localization frame, the steering time delay at a given position is generated from an ideal propagation model and is always assumed to be entirely right. However, the steering time delay to the source position is different from the actual propagation time. Such a difference becomes no more negligible in the outdoor environment and causes a defocus effect, even though the WASN system is well synchronized. (3) Signal passband; when processing the acoustic data collected in outdoor environments, high-pass or band-pass filtering is indispensable because the environmental noise is intense in the low-frequency range, and the source signals in the real world often possess the band-pass characteristic. The synthetic effect of these three factors would make it difficult to achieve stable localization results. The Modified-SRP functional (MSRP) method introduced in [18,19] provides an elegant solution for scalable grids but it is not suitable for band-pass signals. In [21], the authors elaborate on the SRP in band-pass situations and use the GCC-PHAT envelope or frequency-shifted GCC-PHAT to enhance the robustness in such situations. Nevertheless, the above two methods hardly consider the other two factors (the grid and the steering time uncertainty). In [30], the authors propose a Frequency-Sliding GCC (FSGCC) method, which uses singular value decomposition (SVD) or weighted SVD (WSVD) on the FSGCC matrix and can intelligently extract time delay information of the source signal from multiple sub-band GCCs. The authors adopt the WSVD-FSGCC to the MSRP functional for source localization. This solution can provide excellent localization performance in the band-pass situation with scalable grids. However, in outdoor applications, the high computation cost of the SVD of giant matrices is inevitable due to the long GCC range.

Previously, several common acoustic source placements have been proposed in outdoor scenarios. They mostly focus on localizing the source from TDOA [31] and DOA [32,33] measurements. Some uncertainties are then introduced by the estimation error of TDOA or DOA estimating algorithms. Moreover, some useful information is also compressed, which results in unstable performance. In this direction, in this paper, a robust SRP-based outdoor source localization problem is discussed.

In this paper, a modified SRP-based method is proposed, in which the systematic influence of the above inevitable factors in outdoor WASNs scenarios is considered. The localization performance is analyzed using the normalized contribution of the signal components in the SRP function. A sufficient condition dependent on the GCC waveform function for robust on-grid SRP-based source localization is derived by geometrical analysis. The SRP function using GCCs satisfying this condition can suppress the disturbances induced by the grid distance and the uncertain steering time delay. A GCC refinement procedure for band-pass GCCs is then designed, which uses the complex wavelet functions in multiple sub-bands to filter the GCC and averages the envelopes of the filtered GCCs as the equivalent GCC to match the sufficient condition. Simulation results and field experiments demonstrate the excellent performance of the proposed method against the existing SRP-based methods.

The rest of this paper is organized as follows. In Section 2, the outdoor SRP-based source localization problem is formulated. Section 3 gives the sufficient condition in brief and introduces the GCC refinement procedure. The results of the simulation and the field experiment are presented in Section 4. Conclusions are given in Section 5.

## 2. SRP-Based Localization in Outdoor Acoustic Sensor Network

### 2.1. System Models

We discuss the acoustic source localization problem in an *N*-dimensional Euclidean space with *M* distributed microphones (M>N). Let $x\in R^{N}$ be a spatial coordinate vector. Specifically, define $x_{s}$ as the source location and $z_{m}$ as the position of the $m_{th}$ sensor (m=1,2,⋯,M). Let s(t) be the source signal in the time domain, and the received signal of the microphone at $z_{m}$ can be modeled as
(1)$$y_{m}\left[n\right]=\left[h_{m}\left(t\right)*s\left(t\right)+w_{m}\left(t\right)\right]\delta \left(t-n/F_{s}\right),$$
where $h_{m}\left(t\right)$ is the impulse response function representing the propagation of sound from $x_{s}$ to $z_{m}$, the operator “∗” represents the convolution operation, $w_{m}\left(t\right)$ stands for the additive noise signal, and $\delta (t-n/F_{s})$ denotes the sampling process at rate $F_{s}$. When the multi-path delay and non-linear distortion are neglected, the propagation function in the frequency domain can be simplified as
(2)$$H_{m}\left(\omega \right)=A_{m}e^{-j\omega t_{m}},$$
where $A_{m}\in R$ is the amplitude-attenuation factor and $t_{m}$ is the time delay factor. In the frequency domain Equation (Equation 1) can be denoted as
(3)$$Y_{m}\left(\Omega \right)=A_{m}S\left(\Omega \right)e^{-j\Omega F_{s}t_{m}}+W_{m}\left(\Omega \right),$$
where $\Omega =\omega /F_{s}\in \left[-\pi ,\pi \right]$ is the normalized angular frequency, $Y_{m}\left(\Omega \right)$ is the discrete-time Fourier transform (DTFT) of $y_{m}\left[n\right]$, S(Ω) and $W_{m}\left(\Omega \right)$ are the Fourier transforms of s(t) and $w_{m}\left(t\right)$, respectively.

Let $\eta_{m}\left(x\right)\in R$ be the steering time delay function describing the time delay associated with sound propagation from a given location x to $z_{m}$. In practice, it is commonly modeled as the sound traveling time going through the line-of-sight (LOS) path with a constant sound speed $v_{s}$; i.e.,
(4)$$\eta_{m}\left(x\right)=||x-z_{m}\left|\right|/v_{s},$$
where “$∥.∥$” denotes the Euclidean distance. Note that $\eta_{m}\left(x\right)$ is not exactly the sound propagation in reality. Then the SRP function, which is defined as the output power of the filtered-and-sum beam-former, is given by:(5)$$P\left(x\right)=\int_{-\pi }^{\pi }{\left|\sum_{m=1}^{M}G_{m}\left(\Omega \right)Y_{m}\left(\Omega \right)e^{j\Omega_{m}F_{s}\eta_{m}\left(x\right)}\right|}^{2}d\Omega ,$$
where $G_{m}\left(\Omega \right)e^{j\Omega_{m}F_{s}\eta_{m}\left(x\right)}$ is the filter associated with the $m_{th}$ sensor. It can be equivalently expressed in term of GCCs [16]:(6)$$P\left(x\right)=2\pi \sum_{l=1}^{M}\sum_{m=1}^{M}R_{l,m}\left(\eta_{m}\left(x\right)-\eta_{l}\left(x\right)\right),$$
where
(7)$$R_{l,m}\left(\tau \right)=\frac{1}{2\pi }\int_{-\pi }^{\pi }\Psi_{l,m}\left(\Omega \right)Y_{l}\left(\Omega \right)Y_{m}^{*}\left(\Omega \right)e^{j\Omega F_{s}\tau }d\Omega$$
denotes the GCC of the sensor pair {l,m}, τ is the time lag, superscript “${\left(.\right)}^{*}$” represents the conjugate operation, $\Psi_{l,m}\left(\Omega \right)=G_{l}\left(\Omega \right)G_{m}^{*}\left(\Omega \right)$ and denotes the weight function of the associated GCC. Ideally, each $R_{l,m}\left(\tau \right)$ achieves its peak at $\tau =t_{m}-t_{l}$ so that the SRP function is supposed to achieve its maximum value at the source position $x_{s}$, as shown in Figure 1a,b. The Phase Transform (PHAT) weight function
(8)$$\Psi_{l,m}^{PHAT}\left(\Omega \right)=1/\left|Y_{l}\left(\Omega \right)Y_{m}^{*}\left(\Omega \right)\right|$$
is widely used in the TDOA- and SRP-based localization applications. The PHAT-weighted GCC is generally referred to as the GCC-PHAT, and the SRP using the GCC-PHAT is generally referred to as the SRP-PHAT.

Removing those irrelevant and repetitive terms in Equation (Equation 6), the effective component for source localization can be simplified as
(9)$$P_{E}\left(x\right)=\sum_{l=1}^{M-1}\sum_{m=l+1}^{M}R_{l,m}\left(\eta_{m}\left(x\right)-\eta_{l}\left(x\right)\right)=\sum_{p=1}^{C_{M}^{2}}R_{p}\left(\tau_{p}\left(x\right)\right),$$
where *p* is the sequence number of the valid sensor pair $c_{p}=\left\{l,m\right\}\left(l<m\right)$ and is deduced to be p=(2M−l)(l−1)/2+m−l, varying from one to a combinatorial number $C_{M}^{2}$; $\tau_{p}\left(x\right)=\eta_{m}\left(x\right)-\eta_{l}\left(x\right)$ and can be referred to as the steering TDOA function.

### 2.2. Problem Formulation

The classical SRP-based localization method often lacks robustness in outdoor scenarios. The steering time delay function $\eta_{m}\left(x\right)$ in the SRP function is different from the sound propagation in reality denoted as $\eta_{m}^{0}\left(x\right)$, and $\Delta \eta_{m}\left(x\right)=\eta_{m}\left(x\right)-\eta_{m}^{0}\left(x\right)$ is denoted as the steering time-uncertainty function. Similarly, the steering TDOA-uncertainty functions in a pair of sensors can be expressed as
(10)$$\Delta \tau_{p}\left(x\right)\;=\;\Delta \eta_{m}\left(x\right)\;-\;\Delta \eta_{l}\left(x\right)\;=\;\tau_{p}\left(x\right)\;-\;\tau_{p}^{0}\left(x\right),$$
where $\tau_{p}^{0}\left(x\right)\;=\;\eta_{m}^{0}\left(x\right)\;-\;\eta_{l}^{0}\left(x\right)$, representing the real steering TDOA function for a given sensor pair $c_{p}$. This term is usually negligible within a confined space, so it has been rarely discussed in classical SRP models. However, in outdoor applications, the sound propagation is much more unpredictable, resulting in enlarged uncertainty with the increase in distances. The steering time uncertainty can easily be influenced by the geography, temperature, wind, and self-localization error among sensors, and then yields a noticeable defocus effect on the SRP map, as shown in Figure 1c. The GCCs would intersect with each other dispersedly around $x_{s}$.

Since the spatial spectrum generated by the SRP function contains many local extrema and ridged areas, the maximal value of P(x) is usually found through a grid-searching process. Consider a uniform sampling grid (USG) case in $R^{N}$. Define $X_{g}$ as the set of grid points in the candidate searching region ($V\in R^{N}$), and $d_{g}\in R$, $N_{g}\in R$ as the grid distance and the total number of the grids in $X_{g}$, respectively, then the estimated on-grid location is formulated as
(11)$${\hat{x}}_{s}=\arg\max_{x\in X_{g}}P\left(x\right)=\arg\max_{x\in X_{g}}P_{E}\left(x\right).$$

Note that the localization precision depends on the gird resolution. A more accurate estimation usually requires a smaller $d_{g}$. This will leads to a larger $N_{g}$ and significantly increased calculation burden because the number of grids is inversely proportional to the $N_{th}$ power of $d_{g}$ (i.e., $N_{g}\propto {\left(d_{g}\right)}^{-N}$). Hence, the accuracy and feasibility can hardly be balanced in an outdoor WASN system confronting a large search region, for which the minimal grid resolution limited by computing power is much coarser than that in indoor applications. However, most SRP approaches usually work well at subtle grid resolutions, and coarser grid resolution has an undersampled effect, as shown in Figure 1d. The searching process probably would miss the source peak.

It is known that the background noise always dominates at low frequencies in the field environment, and real sound sources often show band-pass characteristics. Thus a band-pass GCC is indeed required. However, the SRP-PHAT with a band-pass source would cause a rippling effect [21], as shown in Figure 1e. The rippling effect does not alter the location of the maximal value of the SRP function. However, it may lead to local extrema and even fake peaks such that the SRP spectrum is susceptible to the two other factors and shows a lack of robustness.

Under the influence of the synthetic effect of the above inevitable factors, the real-world SRP output is illustrated in Figure 1f. It shows that classical SRP implementations hardly deal with all these factors outdoors and yield a divergent localization result.

## 3. A Robust Outdoor SRP-Based Source Localization Method

### 3.1. On-Grid SRP-Based Localization Error Bound Condition

It is known that the SRP-based spatial spectra mainly depend on the phase information of the source components. It is always reasonable to assume that the additive noise of sensors is independent of each other and the source signal, and then it has no spatial preference (which means that they have zero mean in the phase domain). Their contributions to the SRP spectrum can be neglected and not related to the grid resolution and the steering time uncertainty. Therefore, only the contribution of the source signal is considered in analyzing the SRP function. With the terms of additive noise $w_{m}\left(\tau \right)$ neglected, the weight functions $\Psi_{p}\left(\Omega \right)$ of the sensor pair $c_{p}$ usually can be expressed as
(12)$$\Psi_{p}\left(\Omega \right)=B_{p}\Psi_{0}\left(\Omega \right),$$
where $B_{p}\in R$ is an amplitude-scaling factor irrelevant to the frequency, and $\Psi_{0}\left(\Omega \right)\;=\;\Psi_{0}\left(-\Omega \right)\in R$ is a real function irrelevant to sensors. Substituting Equation (Equation 12) into Equation (Equation 7), the GCC $R_{p}\left(\tau \right)$ can be rewritten as
(13)$$\begin{array}{cc} R_{p}\left(\tau \right) & =\frac{B_{p}A_{l}A_{m}}{2\pi }\int_{-\pi }^{\pi }\Psi_{0}\left(\Omega \right)S\left(\Omega \right)S^{*}\left(\Omega \right)e^{j\Omega F_{s}\left(\tau -\tau_{p}^{0}\left(x_{s}\right)\right)}d\Omega \\ \; & =\frac{B_{p}A_{l}A_{m}C_{0}}{2\pi }R_{0}\left(\tau -\tau_{p}^{0}\left(x_{s}\right)\right), \end{array}$$
where $C_{0}=\max\left|\int_{-\pi }^{\pi }\Psi_{0}\left(\Omega \right)S\left(\Omega \right)S^{*}{\left(\Omega \right)}^{j\Omega F_{s}\tau }d\Omega \right|$, and
(14)$$R_{0}\left(\tau \right)=\frac{1}{C_{0}}\int_{-\pi }^{\pi }\Psi_{0}\left(\Omega \right)S\left(\Omega \right)S^{*}\left(\Omega \right)e^{j\Omega F_{s}\tau }d\Omega ,$$
is the amplitude-normalized version of the weighted self-correlation function of the source signal s(t). Hence, each GCC contains the same waveform function $R_{0}\left(\tau \right)$ with different time-shifting factors $\tau_{p}^{0}\left(x_{s}\right)$ and amplitude factors $B_{p}A_{l}A_{m}/C_{0}$. In practice, the range information in amplitude is usually less stable or accurate than in time delay. Thus, a normalized mapping function representing the contribution of the source component in the SRP function can be constructed as
(15)$$F_{E}\left(x,x_{s}\right)=\frac{1}{C_{M}^{2}}\sum_{p=1}^{C_{M}^{2}}R_{0}\left(\tau_{p}\left(x\right)-\tau_{p}^{0}\left(x_{s}\right)\right).$$

In the above equation, the amplitude factors $B_{p}A_{l}A_{m}/C_{0}$ between different sensor pairs are removed. Thus, each pair yields an equal contribution to the SRP function. Note that $F_{E}\left(x\right)\in \left[-1,1\right]$ has a definite value range regardless of the sensor number *M*.

For a given grid distance $d_{g}\in R_{>0}$, an arbitrary uniform sampling grid set in $R^{N}$ can be expressed as
(16)$$X\left(d_{g},x_{g}^{o}\right)=\;\left\{x+x_{g}^{o}:x={\left[n_{1}d_{g},\cdots ,n_{N}d_{g}\right]}^{T};n_{1},\cdots ,n_{N}\in Z\right\},$$
where $x_{g}^{o}\in R^{N}$ is the position of the origin of the set. Then the on-grid location estimation is given by
(17)$$\begin{array}{cc} {\hat{x}}_{s}^{g}\; & =\;\arg\max_{x\in X(d_{g},x_{o})}F_{E}\left(x,x_{s}\right)\\ \; & =\;\arg\max_{x\in X(d_{g},x_{g}^{o})}\frac{1}{C_{M}^{2}}\sum_{p=1}^{C_{M}^{2}}R_{0}\left(\tau_{p}\left(x\right)-\tau_{p}\left(x_{s}\right)+\Delta \tau_{p}\left(x\right)\right). \end{array}$$

It is worth pointing out that the grid resolution, the steering time uncertainty, and band-pass issues are comprehensively considered in the above-simplified SRP function.

The grid issue should be unrelated to the origin position $x_{g}^{o}$. In the real world, the uncertainty functions $\Delta \tau_{p}\left(x\right)$ are hard to closely describe due to many interference factors, and it is reasonable to assume that they have an upper bound $\Delta \tau_{max}$ (i.e., $\left|\Delta \tau_{p}\left(x\right)\right|\leq \Delta \tau_{max}$). $\Delta \tau_{max}$ indicates the steering time delay uncertainty level and can be estimated from the environmental and devices’ conditions. Thus, the robustness of the on-grid localization problem can be described as: given a $d_{g}$ and a $\Delta \tau_{max}$, there exists a ε∈(0,∞) such that
(18)$$∥{\hat{x}}_{s}^{g}-x_{s}∥\leq \varepsilon .$$

Define a level-passed area based on $F_{E}\left(x,x_{s}\right)$:(19)$$M\left(\alpha ,x_{s}\right)≜\left\{x:F_{E}\left(x,x_{s}\right)\geq \alpha \right\}\subseteq R^{N},$$
where α∈R is the level-pass threshold. Then a sufficient condition can be obtained in the following Proposition:

**Proposition** **1.**
*if $M\left(\alpha ,x_{s}\right)\cap X\left(d_{g},x_{g}^{o}\right)\neq \emptyset$ and $M(\alpha ,x_{s})$ is a bounded set (i.e., there exists a $\varepsilon_{M}\in \left(0,\infty \right)$ such that $∥x_{1}-x_{2}∥\leq \varepsilon_{M}$ for all $x_{1},\;x_{2}\in M\left(\alpha ,x_{s}\right)$), then Inequality (Equation 18) is satisfied.*


The proof is given in Section A.1. Thus, the robustness of the on-grid source localization problem can be analyzed in terms of $M\left(\alpha ,x_{s}\right)$.

A practical example of $M\left(\alpha ,x_{s}\right)$ is depicted in Figure 2, and its area shrinks inwards when α increases. The first sub-condition ($M\left(\alpha ,x_{s}\right)\cap X\left(d_{g},x_{g}^{o}\right)\neq \emptyset$) can be satisfied when $M\left(\alpha ,x_{s}\right)$ covers enough areas. The shape of $M(\alpha ,x_{s})$ relates to α, $R_{0}\left(\tau \right)$, $\Delta \tau_{p}\left(x\right)$, and sensor distribution, and it is generally irregular. Consider a closed ball $B^{N}\left(x_{0},r\right)≜\left\{x:|x-x_{0}|\leq r;x_{0},x\in R^{N}\right\}$ with center $x_{0}$ and radius *r*. If
(20)$$r\geq d_{g}\sqrt{N}/2,$$
then $B^{N}\left(x_{0},r\right)\cap X\left(d_{g},x_{g}^{o}\right)\neq \emptyset$ is satisfied. The proof can be seen in Section A.2. Consequently, if $B^{N}\left(x_{s},d_{g}\sqrt{N}/2\right)\subseteq M\left(\alpha ,x_{s}\right)$, then the first sub-condition is satisfied.

Figure 3 illustrates a typical waveform of $R_{0}\left(\tau \right)$, the GCC-PHAT of the passband $\left[{\Omega_{C}-\Omega }_{B},{\Omega_{C}+\Omega }_{B}\right]\;\subset \;\left(0,\pi \right]$, which can be expressed by
(21)$$R_{0}^{PHAT-BP}\left(\tau \right)=\mathrm{sinc}\left(\frac{\Omega_{B}F_{s}}{\pi }\tau \right)\cos\left(\Omega_{C}F_{s}\tau \right).$$

A valid $R_{0}\left(\tau \right)$ is an even and bounded function (i.e., $R_{0}\left(\tau \right)\;=\;R_{0}\left(-\tau \right)$ and $R_{0}\left(\tau \right)\;\in \;\left[-1,1\right]$) and contains a main-lobe around τ=0, where its maximum $a_{m}$ lies. The maximum side-lobe height (or the maximum value outside the main-lobe area if $R_{0}\left(\tau \right)$ has no side-lobes) can be denoted as $a_{s}$, where $a_{s}<a_{m}$.

Let us define a function based on $R_{0}\left(\tau \right)$ by
(22)$$T_{R}\left(a_{T}\right)≜\inf\{|\tau |:R_{0}\left(\tau \right)<a_{T}\},$$
where $a_{T}\in \left[a_{S},a_{M}\right]$ is the level-pass threshold of GCC, “inf{.}” represents the infimum. $T_{R}\left(a_{T}\right)$ represents the half-width of the level-passed section of $R_{0}\left(\tau \right)$ within its main-lobe. It follows that $R_{0}\left(\tau \right)\;\geq \;a_{T}$ if and only if $\tau \;\in \;\left(-T_{R}\left(a_{T}\right),T_{R}\left(a_{T}\right)\right)$.

Based on a geometrical analysis in Section A.3, if $R_{0}\left(\tau \right)$ possesses the following property:(23)$$T_{R}\left(\alpha \right)\geq d_{g}\sqrt{N}/v_{s}+\Delta \tau_{max},$$
then $M\left(\alpha ,x_{s}\right)⊃B^{N}\left(x_{s},d_{g}\sqrt{N}/2\right)$. Therefore, the first sub-condition can be satisfied.

For all α such that $\alpha >\max_{∥x∥\to +\infty }\left\{F_{E}\left(x,x_{s}\right)\right\}$, the second sub-condition ($M(\alpha ,x_{s})$ is a bounded set) is satisfied. The area of $M(\alpha ,x_{s})$ is mainly the superposition of the projection area of the main-lobe sections of GCCs belonging to individual sensor pairs. Denote
$$\Lambda_{p}\left(\tau^{c},T\right)=\left\{x:\right|\tau_{p}\left(x\right)-\tau^{c}\left|\leq T\right\}$$
to be the projection area of the TDOA section $\left[\tau^{c}-T,\tau^{c}+T\right]$ of sensor pair $c_{p}$, where T∈[0,∞) and $\tau^{c}\in \;\left[-\tau_{p}^{max},\tau_{p}^{max}\right]$ are the half-width and the central TDOA of the section, respectively, and $\tau_{p}^{max}=∥z_{l}-z_{m}∥/v_{s}$ is the maximal TDOA value that this sensor pair can produce.

For each sensor pair $c_{p}$, the solution set of the half hyperbolic equation $\tau_{p}\left(x\right)=\tau^{c}$ can be denoted as $\Lambda_{p}\left(\tau^{c},0\right)$ and extends to infinity (i.e., there exists an x such that ∥x∥=∞ and $x\in \Lambda_{p}\left(\tau^{c},0\right)$ ). For two different sensor pairs $c_{i}$ and $c_{j}$, if there exist a $\tau_{i}^{c}\in \left[-\tau_{i}^{max},\tau_{i}^{max}\right]$ and a $\tau_{j}^{c}\in \left[-\tau_{j}^{max},\tau_{j}^{max}\right]$ such that $\Lambda_{i}\left(\tau_{i}^{c},0\right)\subseteq \Lambda_{j}\left(\tau_{j}^{c},0\right)$ or $\Lambda_{i}\left(\tau_{i}^{c},0\right)⊉\Lambda_{j}\left(\tau_{j}^{c},0\right)$, then the half hyperbolic functions $\tau_{i}\left(x\right)=\tau_{i}^{c}$ and $\tau_{j}\left(x\right)=\tau_{j}^{c}$ are not independent. The sense might occur when the sensors of these two pairs are co-linear or have the same axis of symmetry; in the meantime, both $\tau_{i}^{c}$ and $\tau_{j}^{c}$ reach their extremum or become zero. In WASNs, this case rarely happens because the sensor distributions are often irregular. Despite this sense for all sensor pairs, the maximal value of $F_{E}\left(x,x_{s}\right)$ at infinity does not exceed a linear combination of $a_{m}$ and $a_{s}$, which is given as
(24)$$\alpha_{inf}=\frac{C_{N}^{2}a_{m}+\left(C_{M}^{2}-C_{N}^{2}\right)a_{s}}{C_{M}^{2}}.$$

The detailed derivation can be found in Section A.4. If $\alpha >\alpha_{inf}$, then $M(\alpha ,x_{s})$ is bounded.

Combining Inequality (Equation 23) and Equation (Equation 24) together, a sufficient condition for robust on-grid source localization is given by
(25)$$T_{R}\left(\alpha_{inf}\right)\;>d_{g}\sqrt{N}/v_{s}+\Delta \tau_{max}.$$
It means that for a given grid distance $d_{g}$ and steering TDOA uncertainties within $\Delta \tau_{max}$, if the GCC waveform function $R_{0}\left(\tau \right)$ has a wide main-lobe satisfying this condition, then the divergent on-grid location estimation can be avoided.

The SRP-PHAT generates a sharp GCC to increase the TDOA resolution for cases with reverberation or multiple sources. However, as shown in Figure 3, the band-pass effect would bring a narrow main-lobe section and strong side-lobes to the GCC waveform function. It can hardly satisfy the requirement Inequality(Equation 25), which is also shown by the poor performance of SRP-PHAT in Figure 1f. Next, we will introduce a GCC waveform refinement procedure for the band-pass SRP.

### 3.2. Robust SRP-Based Source Localization with Refined GCC Waveform

The condition in Inequality (Equation 25) is too strict for band-pass GCC situations with coarse grid resolution and perceptible steering TDOA uncertainties. Some classical GCC methods utilized low-pass filtering to meet a broader main-lobe requirement, but they are not applicable for band-pass signals. In this section, the GCC is refined to obtain a suitable waveform to modify the SRP function.

Consider a complex wavelet function $\psi_{e}\left(\tau ,\Omega_{C}\right)=u_{e}\left(\tau \right)e^{-j\Omega_{C}F_{s}\tau }$, where $u_{e}\left(\tau \right)\in L^{2}\left(R\right)$ is an even symmetrical function. Applying $\psi_{e}\left(\tau ,\Omega_{C}\right)$ as the filtering function on the GCC-PHAT, the filtered output of $c_{p}$ can be denoted as
(26)$$R_{p}^{CF}\left(\tau ,\Omega_{C}\right)=R_{p}^{PHAT}\left(\tau \right)*\psi_{e}\left(\tau ,\Omega_{C}\right),$$
where $R_{p}^{PHAT}\left(\tau \right)$ is the GCC-PHAT of $c_{p}$.

When the real function $u_{e}\left(\tau \right)$ has an effective support $\left[-\Omega_{B},\Omega_{B}\right]\subset \left[-\pi ,\pi \right]$ in the frequency domain, i.e.,
(27)$$\int_{-\infty }^{\infty }|U_{e}{\left(\Omega \right)|}^{2}d\Omega -\int_{-\Omega_{B}}^{\Omega_{B}}|U_{e}{\left(\Omega \right)|}^{2}d\Omega \ll \int_{-\Omega_{B}}^{\Omega_{B}}{\left|U_{e}\left(\Omega \right)\right|}^{2}d\Omega ,$$
where $U_{e}\left(\Omega \right)$ is the Fourier Transform of $u_{e}\left(\tau \right)$, and if the source is dominant in the frequency band $\left[\Omega_{C}-\Omega_{B},\Omega_{C}+\Omega_{B}\right]\subseteq \left(0,\pi \right]$, then the approximation
(28)$$\begin{array}{cc} R_{p}^{CF}\left(\tau ,\Omega_{C}\right) & =\frac{1}{2\pi }\int_{-\infty }^{\infty }\frac{Y_{p_{l}}\left(\Omega \right)Y_{p_{m}}^{*}\left(\Omega \right)}{|Y_{p_{l}}\left(\Omega \right)Y_{p_{m}}^{*}\left(\Omega \right)|}U_{e}\left(\Omega -\Omega_{C}\right)e^{j\Omega F_{s}\tau }d\Omega \\ \; & \approx \frac{1}{2\pi }\int_{\Omega_{C}-\Omega_{B}}^{\Omega_{C}+\Omega_{B}}\frac{Y_{p_{l}}\left(\Omega \right)Y_{p_{m}}^{*}\left(\Omega \right)}{|Y_{p_{l}}\left(\Omega \right)Y_{p_{m}}^{*}\left(\Omega \right)|}U_{e}\left(\Omega -\Omega_{C}\right)e^{j\Omega F_{s}\tau }d\Omega \\ \; & \approx \frac{1}{2\pi }\int_{\Omega_{C}-\Omega_{B}}^{\Omega_{C}+\Omega_{B}}e^{-j\Omega F_{s}\tau_{p}^{0}\left(x_{s}\right)}U_{e}\left(\Omega -\Omega_{C}\right)e^{j\Omega F_{s}\tau }d\Omega \\ \; & \approx \frac{1}{2\pi }\int_{-\infty }^{\infty }e^{-j\Omega F_{s}\tau_{p}^{0}\left(x_{s}\right)}U_{e}\left(\Omega -\Omega_{C}\right)e^{j\Omega F_{s}\tau }d\Omega \\ \; & =u_{e}\left(\tau -\tau_{p}^{0}\left(x_{s}\right)\right)e^{j\Omega_{C}F_{s}\left(\tau -\tau_{p}^{0}\left(x_{s}\right)\right)} \end{array}$$
exists. It can be observed that the approximate function carries the same envelope as $u_{e}\left(\tau \right)$ and extracts the TDOA information in $\left[\Omega_{C}-\Omega_{B},\Omega_{C}+\Omega_{B}\right]$.

Note that the $R_{p}^{CF}\left(\tau ,\Omega_{C}\right)$ is equal to the time domain approach of the sub-band GCC defined in [30]. Since the main goal is to obtain an equivalent GCC to match the sufficient condition in Inequality (Equation 25), a lightweight approach is to average the envelope of those filtered GCCs of multiple sub-bands in high SNR conditions. According to the power spectral density (PSD) of source signal or other prior knowledge, $N_{q}$ valid sub-bands can be selected with individual central frequency $\Omega_{q}$. The final refined GCC is given by
$$R_{p}^{WR}\left(\tau \right)=\frac{1}{N_{q}}\sum_{q}\left|R_{p}^{CF}\left(\tau ,\Omega_{q}\right)|\approx |u_{e}\left(\tau -\tau_{p}^{0}\left(x_{s}\right)\right)\right|,$$
which has a specific waveform function $R_{0}\left(\tau \right)\approx \left|u_{e}\left(\tau \right)\right|$. Furthermore, the improved spatial function is calculated as
(29)$$P^{WR}\left(x\right)=\frac{1}{C_{M}^{2}}\sum_{p=1}^{C_{M}^{2}}R_{p}^{WR}\left(\tau_{p}\left(x\right)\right)=\frac{1}{C_{M}^{2}N_{q}}\sum_{p=1}^{C_{M}^{2}}\sum_{q=1}^{N_{q}}|R_{p}^{CF}\left(\tau_{p}\left(x\right),\Omega_{q}\right)|.$$

The selection $u_{e}\left(\tau \right)$ has a significant influence on the refinement of GCC. Its envelope $|u_{e}\left(\tau \right)|$ provides the waveform function of refined GCCs. The suitable envelope of a suitable $u_{e}\left(\tau \right)$ should have no side-lobes, i.e., $\left|u_{e}\left(\tau_{1}\right)\right|>\left|u_{e}\left(\tau_{2}\right)\right|\geq 0$ for all $\left|\tau_{1}\right|<\left|\tau_{2}\right|$. Meanwhile, each $U_{e}\left(\Omega -\Omega_{q}\right)$ in the frequency domain serves as a band-pass filter, thus the spectral distribution of $U_{e}\left(\Omega \right)$ should be concentrated to satisfy Inequality (Equation 27). Gaussian function given by
(30)$$u_{e}\left(\tau \right)=e^{-{\left(\Omega_{d}F_{s}\tau \right)}^{2}}$$
which possesses the required properties both in the time domain and in the frequency domain. Then the corresponding complex filtering function $\psi_{e}\left(\tau ,\Omega_{C}\right)$ can be regarded as a complex Morlet wavelet. According to (Equation 25), for a given grid distance $d_{g}$ and steering TDOA uncertainty level $\Delta \tau_{max}$, the parameter $\Omega_{d}$ can be given by
(31)$$\Omega_{d}=v_{s}\sqrt{-\ln\alpha }/\left(F_{s}d_{g}\sqrt{N}+v_{s}\Delta \tau_{max}\right),$$
where *N* is the space dimension, α is the threshold value, which usually can be set as α=0.5. Taking Equation (Equation 31) into Inequality (Equation 27) and dividing (Equation 27) by its right side term, it yields
$$\left[\int_{-\infty }^{\infty }e^{-{\left(\frac{\Omega }{2\Omega_{d}}\right)}^{4}}d\Omega -\int_{-\Omega_{B}}^{\Omega_{B}}e^{-{\left(\frac{\Omega }{2\Omega_{d}}\right)}^{4}}d\Omega \right]/\left(\int_{-\Omega_{B}}^{\Omega_{B}}e^{-{\left(\frac{\Omega }{2\Omega_{d}}\right)}^{4}}d\Omega \right)\ll 1.$$

Thus, the relation of $\Omega_{d}$ and $\Omega_{B}$ can be obtained by the following equivalent equation:$$2\Omega_{d}\left[\int_{0}^{\infty }e^{-\Omega^{4}}d\Omega -\int_{0}^{\Omega_{B}}e^{-\Omega^{4}}d\Omega \right]/\left(\int_{0}^{\Omega_{B}}e^{-\Omega^{4}}d\Omega \right)=c,$$
where *c* is an extremely small number. Then, it can be obtained that
(32)$$\Omega_{B}=2c_{e}\Omega_{d},$$
where $c_{e}$ is the positive solution of the following equation:$$xE_{\frac{3}{4}}\left(x^{4}\right)=\frac{4c}{1+c}\Gamma \left(\frac{5}{4}\right),$$
where $E_{n}\left(x\right)=\int_{1}^{+\infty }\frac{e^{-xt}}{t^{n}}dt,\left(x>0\right)$ and $\Gamma \left(x\right)=\int_{0}^{+\infty }t^{x-1}e^{-1}dt,\left(x>0\right)$. When *c* is set as 0.001(−30 dB), $c_{e}$ in Equation (Equation 32) can be obtained as 2.89.

A simulation is performed to illustrate the effect of the GCC waveform refinement procedure on on-grid SRP-based source localization. As shown in Figure 4, the dot-dashed box shows the range of TDOA within the volume of the nearest gird $x_{g}$, the dashed line with “Δ” shows the real TDOA, which should coincide with the peak of the GCC; the dotted line with “∇” marks $R_{p}\left(\tau_{p}\left(x_{g}\right)\right)$, corresponding to the nearest gird $x_{g}$. The $R_{p}\left(\tau_{p}\left(x_{g}\right)\right)$ of the traditional GCC-PHAT is small, thus leading to poor performance in grid searching. In contrast, the proposed refining method generates a smooth waveform and high values throughout the TDOA region indicated by the box in the figure.

The modified algorithm with the GCC refinement procedure is shown in Algorithm 1, in which ue(τ)=e−(ΩdFsτ)2 is taken as the target waveform function.
**Algorithm 1:** SRP with the waveform refinement procedure**Parameter Setting**(1) Set the maximum steering TDOA error Δτmax=ΔτmaxC+ΔτmaxS, where the sub-items ΔτmaxC and ΔτmaxS are determined by the wind and the synchronization error of sensors, respectively.(2) Set the grid distance dg and searching region V that meet the system requirement. Then the searching grid set Xg is generated.(3) Set the waveform function ue(τ)=e−(ΩdFsτ)2 and α=0.5.(4) Set c=0.001 and compute the bandwidth ΩB using Equation (Equation 32).**Band selecting**(1) Set up the passband ΩL,ΩU(2) Pick up Nq highest PSD bands of the source or divide the passband uniformly.**Source Localization**(1) Calculate the refinement waveform (WR)-SRP function PWR(x) by Equation (Equation 29) at all x∈Xg.(2) Estimate the source location x^s by Equation (Equation 11).

## 4. Experiment Results and Discussion

### 4.1. Numerical Simulations

In this section, we use Monte Carlo simulations to analyze the efficiency of the proposed SRP-based localization method (the SRP functional with the refinement waveform, referred to as WR), compared with the traditional SRP functional with GCC-PHAT (PS), the SRP functional—the envelope of GCC-PHAT (PES) that is designed for acoustic band-pass signals [21], the modified-SRP (M-SRP) functional with GCC-PHAT (PM) [18] in which grid resolution is considered, and the M-SRP functional with the envelope of GCC-PHAT (PEM) in which both band-pass and grid resolution are considered.

In this setup, *M* = 8 sensors and one source are randomly deployed in a monitored area of 200 m by 200 m. The propagation model is set to be the line-of-sight path with a constant sound speed of 345 m/s. The input GCCs are generated by the waveform function in Equation (Equation 21) with passband of $\left[0.15\pi ,0.4\pi \right]$. The steering TDOA uncertainty $\Delta \tau_{p}\left(x\right)$ uniformly distributes over $\left[-\Delta \tau_{max},\Delta \tau_{max}\right]$, where $\Delta \tau_{max}$ is the maximal time uncertainty dependent on the sound-propagation model error and the synchronization error.

We consider four different conditions in WASNs to test the algorithms: (a) a small steering TDOA uncertainty and small grid distance (STSG) condition with $\Delta \tau_{max}=0.1$ ms, $d_{g}=0.1$ m, (b) a large steering TDOA uncertainty and small grid distance (LTSG) condition with $\Delta \tau_{max}=100$ ms, $d_{g}=0.1$ m, (c) a small steering TDOA uncertainty and large grid distance (STLG) condition with $\Delta \tau_{max}=0.1$ ms, $d_{g}=10$ m, (d) a large steering TDOA uncertainty and large grid distance (LTLG) condition with $\Delta \tau_{max}=100$ ms or $d_{g}=10$ m.

The mean absolute error (MAE) $E\left\{∥{\hat{x}}_{s}-x_{s}∥\right\}$ of distance and the cumulative distribution function (CDF) of estimation errors of relative distance are calculated to evaluate the accuracy and robustness of these algorithms, where the relative distance in the cumulative distribution function (CDF) is normalized by the grid distance, i.e.,
(33)$$F\left(e_{u}\right)=P\left\{∥{\hat{x}}_{s}-x_{s}∥/d_{g}\leq e_{u}\right\},$$
where $e_{u}$ is the relative positioning error that is determined as the system requirement. Specifically, the 95th percentile of the localization error in meters is computed as $F^{-1}\left(0.95\right)\cdot d_{g}$.

The MAE and 95th percentile results are listed in Table 1. All the localization algorithms can obtain the best estimation accuracy in the STSD condition in which the defocus effect and undersampled effect are slight. When the steering TDOA uncertainty or the grid distance increases, the MAE would increase. However, compared with the PS, PES, PM, and PEM methods, the MAE in the WR has almost the smallest estimate error because all these factors have been considered. The 95th percentile has similar results with the MAE, which indicates that the proposed WR method has a stable localization performance in outdoor conditions.

Figure 5a–d depict the CDF of each algorithm in the range $e_{u}\in \left[0.5,100\;m/d_{g}\right]$ under the four conditions. Specifically, the CDF curves will increase rapidly with the location error in the fine condition, and then the estimate errors are the smallest for all the algorithms in the STSG. The CDF curve will move down as the grid distance $d_{g}$ and steering TDOA uncertainty $\Delta \tau_{max}$ increase, such as in the LTSG, STLG, and LTLG. Since the steering TDOA uncertainty is not considered in PES and PEM, their descent range of CDF in the SDLG is lower than that in the LDSG. Among these localization algorithms, the CDF of the WR is the highest or very close to the highest (STLG), and the PEM method is better than the PS, PES, and PM. The proposed WR method is very robust even though the condition becomes abominable.

Furthermore, Figure 6 presents the MAE in four situations: (a) fixed small steering TDOA uncertainty (ST) with $\Delta \tau_{max}$ = 0.1 ms, $d_{g}$ ranges from 0.1 m to 50 m; (b) fixed large steering TDOA uncertainty level (LT) with $\Delta \tau_{max}$ = 100 ms, $d_{g}$ ranges from 0.1 m to 50 m; (c) fixed small grid distance (SG) with $d_{g}$ = 0.1 m, $\Delta \tau_{max}$ range from 0.1 ms to 100 ms; (d) fixed large grid distance (LG) with $d_{g}$ = 10 m, $\Delta \tau_{max}$ range from 0.1 ms to 100 ms. The MAE increases with $d_{g}$ or $\Delta \tau_{max}$ significantly, and this indicates that the steering TDOA uncertainty and grid distance have a severe influence on the performance of source localization. In each situation, the PS and PM produce larger MAE than the other algorithms when $d_{g}$ and $\Delta \tau_{max}$ are small because they are not applied to band-pass signals. Since the scalable grid sampling and steering TDOA uncertainty are not considered in the PES, it shows reliable performance only when $d_{g}\leq 1$ m and $\Delta \tau_{max}\leq 1$ ms. The PEM considered both grid size and band-pass effect; thus, it achieves the best performance in the small $\Delta \tau_{max}$ case. However, the MAE becomes worse when the influence caused by the steering TDOA uncertainties is more significant than by the grid size. The WR obtains the MAE close to the PEM when $\Delta \tau_{max}$ is small. Moreover, it is the smallest in all the other situations. These results abundantly demonstrate its excellent robust performance.

### 4.2. Field Experiment

In this experiment, seven nodes are distributed in a park, as shown in Figure 7a,b. Each node consists of a microphone sensor, a Wi-Fi module, and a GPS module for self-localization and time calibration. The monitoring area has the same 200 m × 200 m in addition with a hillock. A portable speaker generates the sound signals at 12 positions inside the area, such as the Gaussian signal (S-G), the whistle of vehicles (S-V) representing an urban source, and birdsong (S-B) representing a field source. The temperature was approximately 30 °C, and the wind speed is slower than 3 m/s. Therefore, in the proposed method $\Delta \tau_{max}$ can be set to be 10 ms fully considering the self-localization error of the sensors and the effect of wind.

The sampling frequency is 10,000 Hz and Figure 7c shows the PSDs of both the background noise and received source signals, which are obtained with the Burg method of 50 order number and 2048 FFT length. The PSDs of the source signals are collected at about 30 m away from the speaker. Because the environmental noise is mainly distributed in the frequency bands below 1500 Hz, the passband is set to be (1500 Hz, 3500 Hz) for all sources. The estimated SNRs are shown in Figure 7d, and the SNRs of the full band (0, 5000 Hz) and of the passband (1500 Hz, 3500 Hz) are plotted in solid lines and dashed lines, respectively. For the three source types, the SNR is improved by 20 dB∼30 dB.

The recorded data are divided into 1242 two-second audio frames. SRP algorithms with full-band and band-pass cross-correlation (referred to as CSF and CSB) are added to analyze the necessity of band-pass signals. The PS and PM are not included since they have been proven unreliable in the simulation. Then the candidate SRP-based locators compared in this sub-section include: (1) SRP with full-band GCC (CSF), (2) SRP with band-pass GCC (CSB), (3) SRP with the envelope of band-pass GCC-PHAT (PES), (4) MSRP with the envelope of band-pass GCC-PHAT (PEM) and (5) WR-SRP with band-pass GCC (WR). A well known TDOA-based localization method [13] (referred to as TC) is also compared as a reference in which the TDOAs are obtained by band-pass GCC-PHATs.

The MAE and the 95th percentile of the localization errors of the TC method and the SRP-based methods with different grid distances ($d_{g}\in \left\{0.1,1,10\right\}$ m) are listed in Table 2. Moreover the MAEs with grid distance $d_{g}$ ranging from 0.1 m to 50 m are presented in Figure 8a. Figure 8b–d give the CDF curves at the three grid distances ($d_{g}\in \left\{0.1,1,10\right\}$ m).

Like the simulation, the MAEs increase and the CDF curves move down as the grid distance increases. The MAE of the TC method is the highest because some sensor pairs might produce very severe TDOA measurements in noisy acoustic environments. Its CDF curve also shows that the solution is not stable. By comparing the result of CSF and CSB, the band-pass GCC can significantly enhance the SNR and the localization performance. The PES and PEM obtain more significant localization errors and lack robustness, which indicates the influence of the steering TDOA uncertainty is very remarkable. The proposed WR method achieves the best estimation for all the grid distances, which thoroughly verifies its effectiveness.

## 5. Conclusions

In this work, a novel and robust Steered Response Power (SRP)-based source localization approach is proposed to localize the band-pass source in outdoor WASNs with steering time delay uncertainty and coarser spatial grids. The robustness of on-grid source localization is analyzed by a sufficient condition, in which the relation between GCC signal waveform and on-grid localization error is demonstrated. A band-pass GCC refinement procedure is designed to meet the sufficient condition for enhancing the on-grid source localization performance. The Monte Carlo simulation and field experiment show that the proposed method has a robust performance in outdoor WASNs scenarios, compared with some state-of-the-art SRP-based methods.

## Figures and Tables

**Figure 1 sensors-21-01591-f001:**
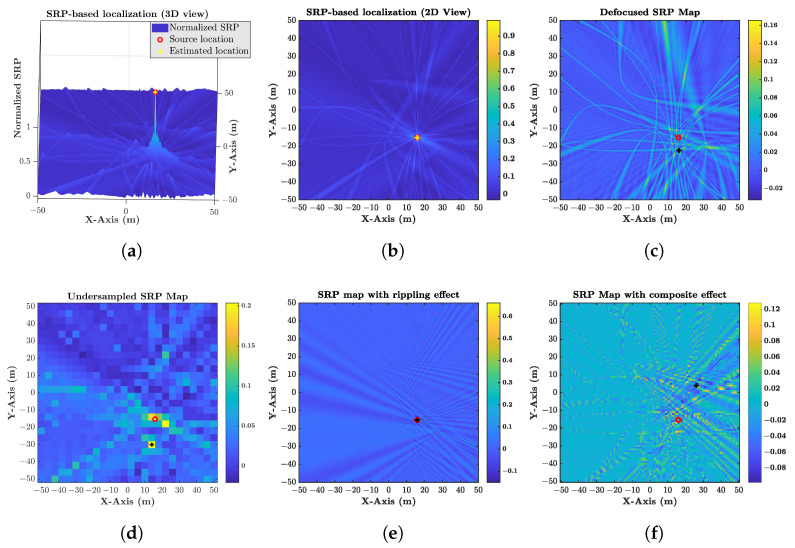
Comparison of the ideal steered response power (SRP)-based source localization in an ideal case and with the unexpected effects (the symbols “o” and “+” represent the source position and the estimated position, respectively): (**a**) SRP map (3D view); (**b**) Ideal SRP map (2D view); (**c**) defocus effect from steering time uncertainties; (**d**) undersampled effect from coarse grid; (**e**) rippling effect from band-pass generalized cross-correlations (GCCs); (**f**) combined effect.

**Figure 2 sensors-21-01591-f002:**
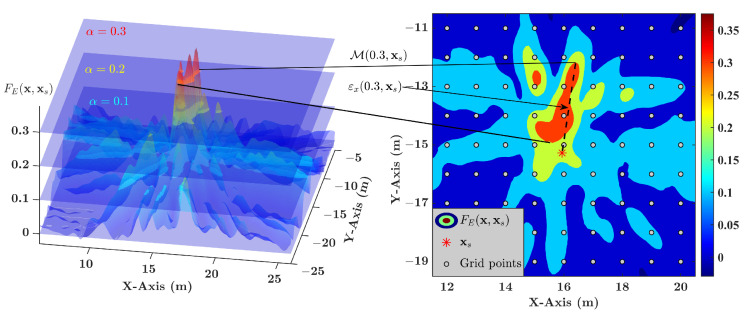
Illustration of the level-pass area $M\left(\alpha ,x_{s}\right)$. (Orange: $M\left(0.3,x_{s}\right)$; yellow green: $M\left(0.2,x_{s}\right)$; celeste: $M\left(0.1,x_{s}\right)$).

**Figure 3 sensors-21-01591-f003:**
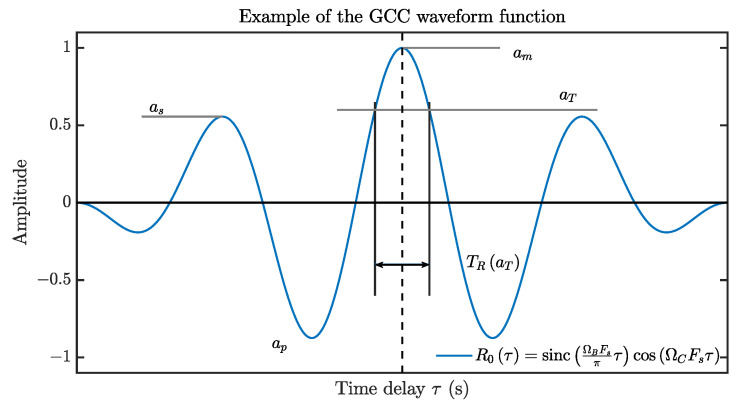
An example of $R_{0}\left(\tau \right)$.

**Figure 4 sensors-21-01591-f004:**
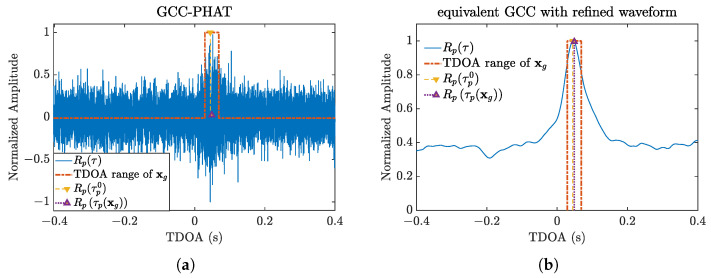
An example of refined GCC from field data: (**a**) GCC-Phase Transform (PHAT); (**b**) refined GCC.

**Figure 5 sensors-21-01591-f005:**
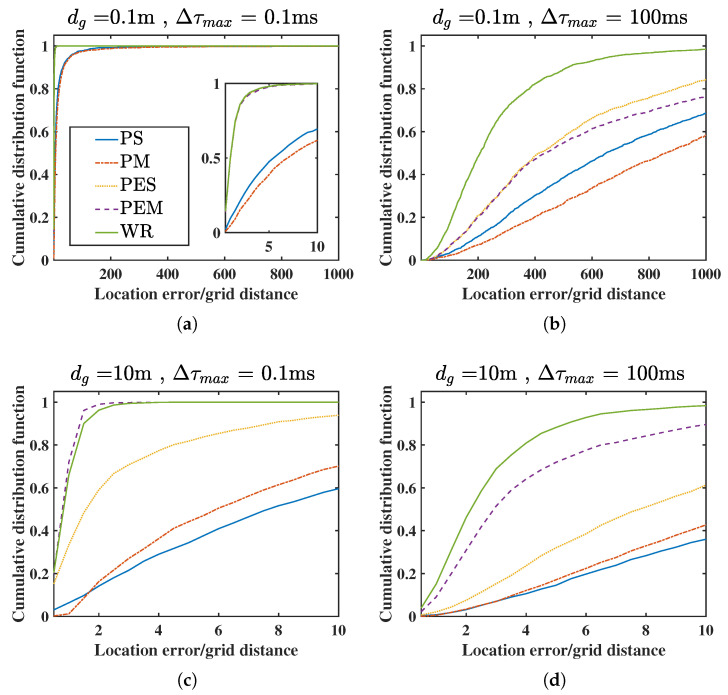
Simulation comparison in the cumulative distribution function (CDF) of relative distance error. (**a**) small steering time difference of arrival (TDOA) uncertainty and small grid distance (STSG); (**b**) large steering TDOA uncertainty and small grid distance (LTSG); (**c**) small steering TDOA uncertainty and large grid distance (STLG); (**d**) large steering TDOA uncertainty and large grid distance (LTLG).

**Figure 6 sensors-21-01591-f006:**
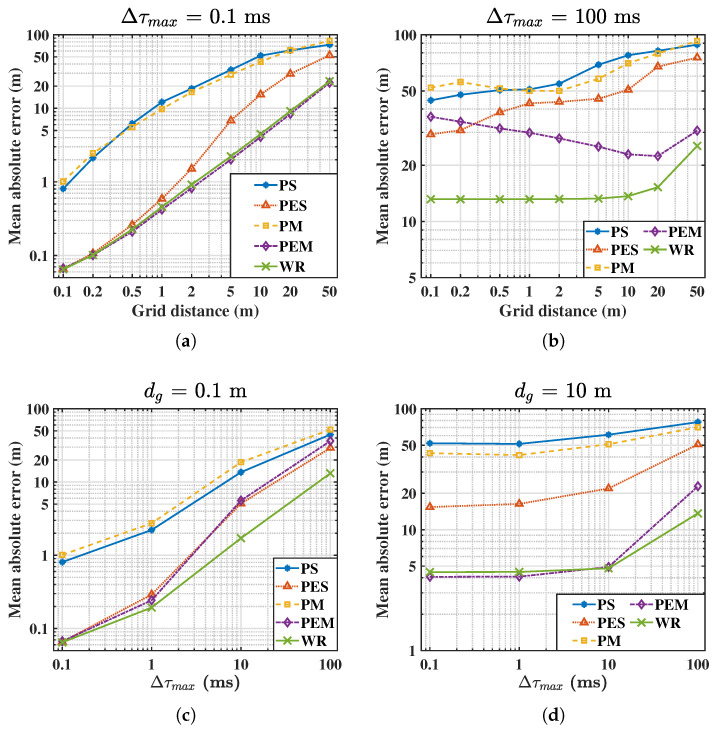
The mean absolute errors (MAEs) under different conditions. (**a**) small steering TDOA uncertainty (ST) ($\Delta \tau_{max}$ = 0.1 ms, $d_{g}\in [0.1\;$m,50m]); (**b**) large steering TDOA uncertainty level (LT) ($\Delta \tau_{max}$ = 100 ms, $d_{g}\in [0.1$ m,50m]); (**c**) small grid distance (SG) ($d_{g}$ = 0.1 m, $\Delta \tau_{max}\in [0.1\;$ms,100ms]); (**d**) large grid distance (LG) ($d_{g}$ = 10 m, $\Delta \tau_{max}\in$(0.1 ms,100 ms)).

**Figure 7 sensors-21-01591-f007:**
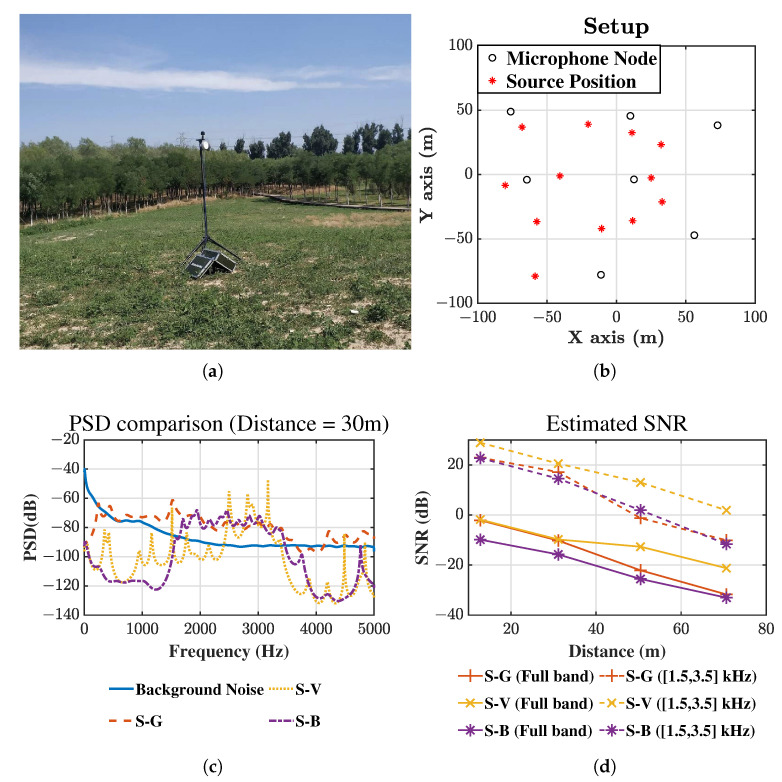
Setup of the field experiment (**a**) Device. (**b**) Distribution. (**c**) Estimated power spectrum density of sensor signal 30 m away from source. (**d**) Estimated signal to noise ratio.

**Figure 8 sensors-21-01591-f008:**
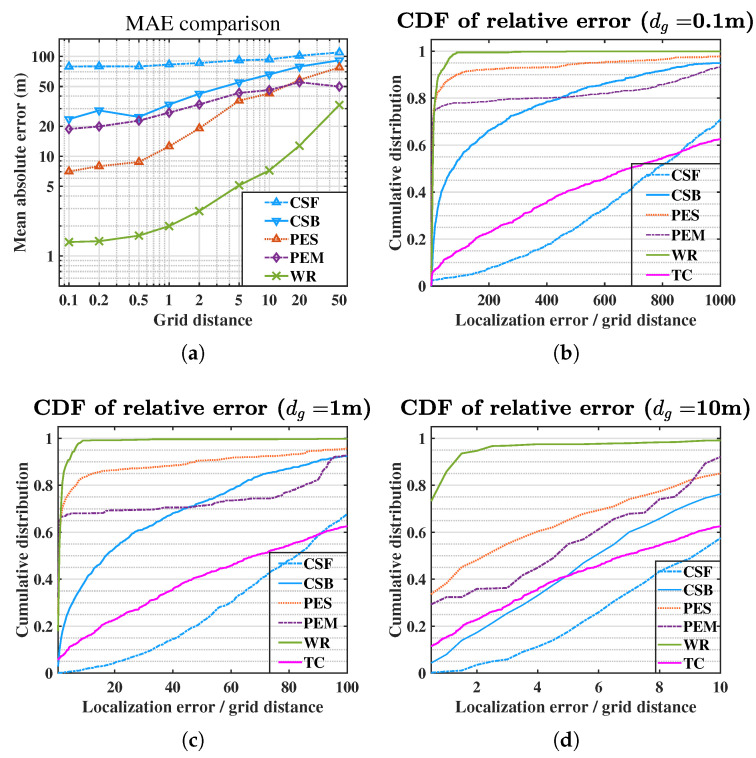
Experiment results: (**a**) MAE comparison; (**b**) CDF of relative error at $d_{g}=0.1$ m; (**c**) CDF of relative error at $d_{g}=1$ m; (**d**) CDF of relative error at $d_{g}=10$ m.

**Table 1 sensors-21-01591-t001:** Mean absolute error (MAE) and 95th percentile under different conditions in the simulation.

MAE (m)					
Condition	PS	PES	PM	PEM	WR
STSG	0.81	0.07	1.01	0.07	0.06
LTSG	44.53	29.27	52.04	36.37	13.16
STLG	51.90	15.39	42.97	4.07	4.46
LTLG	77.64	50.74	70.37	22.88	13.65
95th percentile (m)					
Condition	PS	PES	PM	PEM	WR
STSG	2.83	0.17	2.99	0.18	0.17
LTSG	123.13	82.61	128.10	118.61	33.43
STLG	147.04	58.81	124.39	7.11	9.24
LTLG	172.37	139.73	163.95	74.07	34.68

**Table 2 sensors-21-01591-t002:** Mean absolute error (MAE) and 95th percentile under different conditions in the field experiment.

MAE (m)						
Condition	TC	CSF	CSB	PES	PEM	WR
no grid	102.2	-	-	-	-	-
$d_{g}$ = 0.1 m	-	79.2	23.5	7.1	18.7	1.4
$d_{g}$ = 1 m	-	83.0	33.0	12.6	27.4	2.0
$d_{g}$ = 10 m	-	93.3	66.0	42.6	46.1	7.2
95th percentile (m)						
Condition	TC	CSF	CSB	PES	PEM	WR
no grid	322.8	-	-	-	-	-
$d_{g}$ = 0.1 m	-	146.5	100.8	53.7	105.0	5.4
$d_{g}$ = 1 m	-	150.4	113.1	91.6	105.1	6.0
$d_{g}$ = 10 m	-	171.8	149.0	138.5	104.6	21.0

## Data Availability

Publicly available datasets were analyzed in this study. This data can be found here: https://1drv.ms/u/s!AskSoQGpB3VUgfIqsxtYhosVrGyzOg?e=pnfutC.

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
