# Peer review of "A Robust Steered Response Power Localization Method for Wireless Acoustic Sensor Networks in an Outdoor Environment"

_sensors, 2021, doi:10.3390/s21051591_

Round 1
Reviewer 1 Report
This paper considers a robust steered response power outdoor source localization problem, some severe factors like steering time delay and band-pass source signal are involved to enhance the estimation performance. This topic is important and the idea is interesting in the field of SRP localization, a novel GCC refinement method is proposed for the robustness of localization.
The following revisions are suggest for the authors. 1. The sufficient condition dependent on spatial grid and steering time uncertainty is not clear, the authors should explain the relation between these factors. 2. In the introduction, should highlight the differences among their method and the existing works. 3. Some statements are confused, for example eq.(19), maybe it is maybe suitable to represented by Lemma or theorem. And in the eq(22), the statement is also confused, is not strictly correspond with the result. The author should need to revise this part. 4. Eq.(27) is approximated by some conditions, but it is not given in the paper. 5. Eq.(28) is also further explained for the approximation.Author Response
Please see the attachment.

Reviewer 2 Report
The authors present an interesting and elaborate study on Steered-Response-Power (SRP) methods for sound localization with band-pass acoustic signals. They propose a GCC enhancement method using complex wavelet functions in multiple sub-bands to filter the GCCs, averaging output envelopes to match a sufficient condition. The authors describe clearly the problems arising in the covered scenario and show meaningful results with simulated experiments and real-world recordings.
In general, the paper is technically sound and the results are convincing, although the authors should carefully revise the manuscript in terms of English writing.
As the paper is focused on band-pass acoustic localization, I would highly recommend the authors to relate their contribution or compare their method to other recent approaches in the field that also consider sub-band GCC processors:
M. Cobos, F. Antonacci, L. Comanducci and A. Sarti, "Frequency-Sliding Generalized Cross-Correlation: A Sub-Band Time Delay Estimation Approach," in IEEE/ACM Transactions on Audio, Speech, and Language Processing, vol. 28, pp. 1270-1281, 2020, doi: 10.1109/TASLP.2020.2983589.
